# Comparison of the ColiPlate™ Kit with Two Common *E. coli* Enumeration Methods for Water

**Cassi J. Gibson, Abraham K. Maritim and Jason W. Marion \***

Department of Environmental Health Science, Eastern Kentucky University, Richmond, KY 40475, USA;
cassi_gibson40@mymail.eku.edu (C.J.G.); abraham_maritim1@mymail.eku.edu (A.K.M.)
\* Correspondence: jason.marion@eku.edu; Tel.: +1-859-622-6343

**Abstract:** Quantitatively assessing fecal indicator bacteria in drinking water from limited resource settings (e.g., disasters, remote areas) can inform public health strategies for reducing waterborne illnesses. This study aimed to compare two common approaches for quantifying *Escherichia coli* (*E. coli*) density in natural water versus the ColiPlate™ kit approach. For comparing methods, 41 field samples from natural water sources in Kentucky (USA) were collected. *E.'coli* densities were then determined by (1) membrane filtration in conjunction with modified membrane-thermotolerant *E. coli* (mTEC) agar, (2) Idexx Quanti-Tray® 2000 with the Colilert® substrate, and (3) the Bluewater Biosciences ColiPlate kit. Significant correlations were observed between *E. coli* density data for all three methods ($p < 0.001$). Paired *t*-test results showed no difference in *E. coli* densities determined by all the methods ($p > 0.05$). Upon assigning modified mTEC as the reference method for determining the World Health Organization-assigned "very high-risk" levels of fecal contamination (>100 *E. coli* CFU/100 mL), both ColiPlate and Colilert exhibited excellent discrimination for screening very high-risk levels according to the area under the receiver operating characteristic curve (~89%). These data suggest ColiPlate continues to be an effective monitoring tool for quantifying *E. coli* density and characterizing fecal contamination risks from water.

**Keywords:** fecal indicator bacteria; water quality; *E. coli*; ColiPlate; Colilert; environmental microbiology





## 1. Introduction

*Escherichia coli* (*E. coli*) are routinely used as fecal indicator bacteria for characterizing human infectious disease risk following exposure to drinking and recreational waters [1–4]. In drinking water, *E. coli* are preferred over other indicators due to their sensitivity for detecting fecal contamination and their ability to be simply and inexpensively enumerated [5,6]. Specifically, for drinking water, *E. coli* detection and/or enumeration are valued for assessing treatment effectiveness as part of routine monitoring protocols aimed at protecting public health [6]. Meta-analyses [7,8] and latter individual studies [9,10] have also supported the utility of culturable *E. coli* for predicting human illness risk among recreational water users. Accordingly, *E. coli* density is routinely evaluated at bathing areas per the European Union's and US bathing/recreational water quality directives [11,12].

Associations between *E. coli* density and gastrointestinal (GI) illnesses in drinking water studies from low-resource settings have had varied findings [13]. The broad group of total and fecal coliform bacteria do not appear to be associated with GI illnesses in household drinking water studies [14]; however, a meta-analysis of 14 studies using specifically *E. coli* demonstrated a positive relationship between diarrhea occurrence and *E. coli* density [15]. Some studies, such as by Profitós et al. [16], failed to show an association between *E. coli* density and GI illness frequency, which they suggested might have been due to a small sample size of 120 households ($n = 785$ individuals). In a large study across 50 rural communities in Bangladesh (>12,000 monthly observations) using a robust cohort study design, *E. coli* density was positively associated with diarrhea [17]. Among several

reasons, Levy et al. [13,18] notes that in smaller studies, sample sizes may be too small to see significant associations.

The methods for enumerating *E. coli* vary across studies attempting to understand relationships between microbiological water quality and gastrointestinal illness risk. Methods requiring refrigerated media, incubators, heavy batteries, direct electricity, and/or relatively significant costs may not be practical in resource-limited settings. These logistical arrangements can increase study costs and labor resources thereby decreasing sample size. Accordingly, innovative approaches and previously studied methods, such as low-cost field-deployable hydrogen sulfide testing, have recently been reconsidered and assessed in these settings, such as following an earthquake in Haiti [19]. Overall, scientific consensus remains that there is tremendous need for water quality tests that can practically assess fecal contamination in low-resource settings [18–21]. Access to reliable and practical methods could mitigate the sample size challenges described by Levy et al. in epidemiological studies while also benefiting communities [13,17].

One approach to enumerating *E. coli* without requiring refrigerated media or extensive laboratory resources is the ColiPlate™ kit (Bluewater Biosciences, Mississauga, ON, Canada). ColiPlate uses a 96-well microplate format and has several similarities to Idexx Colilert (Idexx Laboratories, Westbrook, ME, USA). The advantages of ColiPlate relate to the simplicity of the approach and affordability. One hundred-milliliter samples can be poured into ColiPlate or pipetted, preferably with a multichannel pipette. ColiPlate contains media embedded into the 96 wells and does not require any mixing or separate transport of media. Colilert, like ColiPlate, is shelf-stable, but requires the use of specialized molded plastic, foil-backed Idexx Quanti-Trays and either an Idexx Quanti-Tray Model 2X Sealer (20 kg) or the newer Quanti-Tray Sealer Plus (11 kg). Due to the requirement of a tray sealer, Idexx has a relatively high cost per sample unless processing a high volume of samples. Additionally, tray sealers have specific electrical requirements that may not be upheld in low-resource settings. Whereas incubation should occur for both methods, in low-resource settings, improvisation can occur as there are low-cost incubator solutions described in the literature, e.g., use of heated water [22].

There are similarities between the Colilert and ColiPlate methods, particularly in terms of incubation time (24 h) and temperature (35–37 °C), ability to simultaneously enumerate total coliforms and *E. coli*, utilization of the most probable number (MPN) approach with MPN tables, and the utilization of a common substrate for detecting *E. coli*.

ColiPlate and Colilert both utilize defined substrate technology (DST) for enumerating total coliforms and *E. coli*. Both screen for the enzyme β-D-galactosidase which is a common enzyme in total coliform bacteria. Both also screen for the activity of β-D-glucuronidase which is an enzyme commonly observed in *E. coli*. Colilert and ColiPlate differ in their total coliform detection. Colilert utilizes *o*-nitrophenyl-β-D-galactopyranoside (ONPG) for detecting and subsequently quantifying total coliform-positive wells by eliciting a visible yellow color change due to β-D-galactosidase activity on ONPG [23]. ColiPlate detects total coliform-positive wells by generating a blue color change due to β-D-galactosidase activity on X-Gal (also known as BCIG or 5-bromo-4-chloro-3-indolyl-β-D-galactopyranoside) [24].

In terms of enumerating *E. coli*, the approaches of Colilert and ColiPlate are similar. Wells that are positive for total coliforms by either method can be immediately analyzed for *E. coli* by viewing a 96-well ColiPlate or Idexx Quanti-Tray under a longwave UV light to identify fluorescing (glowing) blue wells, thereby demonstrating glucuronidase activity by, presumably, *E. coli*. Specifically, ColiPlate and Colilert utilize the fluorescent chemical 4-methylumbelliferyl-β-D-glucuronide (MUG) for detecting *E. coli* [23,24].

Given the potential needs associated with the use of ColiPlate kits in low-resource settings and/or by citizen scientists, this evaluation was performed. Citizen scientists are needed for making substantial contributions to the United Nations Sustainable Development Goals, and for doing so, practical approaches and tools are needed [25]. To date, peer-reviewed studies evaluating the ColiPlate versus membrane filtration (MF) approaches, enzyme substrate methods, or other methods are limited to primarily one

study [26] despite substantial use of ColiPlate kits by community, academic, and government entities [16,27–29] and recognition in the catalog of microbial drinking water tests for limited resource settings [30]. Due to the limited assessment of ColiPlate in comparison to other *E. coli* enumeration methods, we explored associations between the densities determined by ColiPlate, Colilert, and modified membrane-thermotolerant *E. coli* (mTEC) agar methods, hypothesizing that all would be positively associated.

## 2. Materials and Methods

### 2.1. Sample Collection and Analysis Preparation

A total of 41 natural water samples were collected from natural waters in the south-central region of Kentucky in the United States from 24 different bodies of water. Overall, 23 samples were collected from lotic waters and 18 samples were from lentic environments. All the samples were collected in August and September 2016 from within the Kentucky River basin. Specifically, 17 samples were from medium-sized streams (Taylor Fork, Silver Creek, Ballard Branch, Hickman Creek, Otter Creek, and Frozen Creek); six samples were from the mainstem of the medium-sized Kentucky River. A total of 12 samples were obtained from the shoreline and the fishing dock of a small lake (Lake Reba [0.32 km$^2$]), and six samples were obtained from agricultural (farm) ponds. The six Kentucky River sample locations are impaired solely for mercury/fish bioaccumulation. The majority of the other locations are impacted by nutrient enrichment from agriculture (pastureland and/or row crop agriculture) and thus impaired for aquatic life by not meeting the respective warmwater habitat designations for the locations required to be assessed by the Commonwealth of Kentucky.

All the samples were collected in sterile (autoclaved) 1.5-L Nalgene bottles by slowly sweeping the bottles under the surface without disturbing the sediments. All the samples were from the shoreline except six dock samples at Lake Reba. All the samples were evaluated within four hours of collection using three methods for each sample to enable comparison of results by the method type. The single sample container for each collection location obtained a sufficient volume of water to enable all the three methods to be performed.

In the laboratory, supervised analyses were performed by upper-division university and graduate student investigators who performed all the three analyses to simulate method use by citizen scientists. Two most probable number (MPN) methods (ColiPlate and Colilert) were used for enumerating *E. coli* density in accordance with the manufacturer's instructions for each method. Prior to initiating each method, the samples were mixed by gentle shaking.

### 2.2. ColiPlate Analysis

For the ColiPlate analysis, the direct pour method according to the manufacturer's instructions was used. Specifically, sample water was poured into a media-containing 96-well plate. The 96-well plates were then incubated at 35 °C for 24 h alongside Colilert Quanti-Trays. Following incubation, the ColiPlate wells colored blue under visible light were deemed total coliform-positive wells. Among these total coliform wells, those that fluoresced or glowed in the 365-nm longwave UV viewing cabinet (Spectroline, Westbury, NY, USA) were enumerated as *E. coli*-positive wells. Based upon the enumeration of positive wells for each method, the ColiPlate MPN table [31] was used to determine *E. coli* density per 100 mL. The detection range was 3–2424 MPN per 100 mL. In addition to these methods, one negative control using an autoclaved 1.5-L bottle containing autoclaved distilled water was performed.

### 2.3. Colilert Analysis

For the Colilert analysis, we poured 100 mL of each sample into a sterile plastic sampling vial. The Colilert medium was then added, and the sample vessel was inverted 40 times to mix and dissolve the medium. The mixed solution was then placed into an

Idexx Quanti-Tray 2000™ sealed using the Quanti-Tray sealer and then incubated for 24 h at 35 °C. Quanti-Tray 2000 has 49 large wells and 48 small wells with a quantification range of 1–2419.6 MPN per 100 mL [32].

Following incubation, *E. coli* were enumerated. First, total coliform-positive wells (yellow/gold in color) were identified and enumerated from the 49 large and 48 small Idexx Quanti-Tray 2000 wells. Then, the Quanti-Tray was placed in a 365-nm Longwave UV viewing cabinet, and the yellow/gold wells were assessed. The yellow/gold wells that fluoresced or glowed blue in the UV viewing cabinet were counted as *E. coli*-positive wells. The number of large wells and small wells were then compared to the Colilert MPN table for Quanti-Tray 2000 [32].

### 2.4. E. coli Enumeration by Membrane Filtration with Modified mTEC Agar

The colony-forming unit (CFU) method for enumerating *E. coli* density was performed using a membrane filtration method with modified mTEC agar plates (Aquaplates, Perkinsville, VT, USA) in accordance with EPA Method 1603 [33]. Filtration volumes of 20 mL, 50 mL, and 100 mL were used for all the samples using water from the 1.5-L sample bottle. The samples were incubated at 35 °C for 2 h and then placed into an incubator at 44.5 °C for 22 h.

Following incubation, magenta- or red-colored colonies were counted as *E. coli*. The results for the 20-mL and 50-mL sample volumes were converted to CFU per 100 mL. Among the three plates associated with the three filtration volumes, plates with 20–80 magenta or red colonies were used for determining the reported CFU per 100 mL. If none of the three plates had fewer than 80 colonies, then the 20-mL filtration volume was used for determining CFU per 100 mL. If multiple plates had an acceptable amount of *E. coli* colonies, the average number of CFU per 100 mL was used.

### 2.5. Statistical Analysis

Stata 15 (StataCorp, College Station, TX, USA) was used for all the statistical analyses. Given that the distributions of the *E. coli* data for each of the three methods were all skewed and not normally distributed (Shapiro–Wilk test $p < 0.00001$), the data were transformed using $log_{10}$ transformation which normalized the data for all the three variables (Shapiro–Wilk test $p > 0.05$). Among the 41 water samples, one sample exceeded the range of all the tests and was excluded. Summary statistics (mean, median, and range) were obtained for the 40 remaining samples, and Pearson correlation analysis was performed to assess the strength of association between the three methods for comparison with other studies providing such data. Spearman's rank correlation analysis was also performed to enable comparison with other studies. A box-and-whisker plot and scatterplots using the log-transformed data were used to visually explore the relationships between the two MPN methods with the CFU method. To assess any significant differences between the various methods, *t*-tests using the log-transformed data were performed.

Given that the World Health Organization's water quality guidelines indicate that *E. coli* density exceeding 100 CFU or MPN per 100 mL presents a very high risk to potential drinking water users [34], the two MPN methods (ColiPlate and Colilert) were also evaluated for their sensitivity (the ability to correctly classify a positive result) and specificity (the ability to correctly classify a negative result) for detecting *E. coli* levels exceeding 100 CFU per 100 mL. To enable this evaluation, *E. coli* densities obtained by ColiPlate and Colilert methods were evaluated against an MF method using modified mTEC agar for detecting 100 CFU per 100 mL. The MF method with modified mTEC agar was set as the reference method for several reasons, specifically because (1) an outgroup was needed for making comparisons, (2) Colilert and ColiPlate have similarities as MPN methods and for the stability of their media in environments without refrigeration, (3) the MF method with modified mTEC agar has been used for deriving human health criteria for recreational water exposure [12,35], and (4) this MF method is a US EPA method approved for nationwide (US) use. The use of an MF method for this study as a statistical reference

(or outgroup) does not imply that it is more correct in the enumeration of the true *E. coli* density than the other methods, and it is noteworthy that the Idexx Colilert method is also US EPA-approved, more widely used worldwide, and is included in Standard Methods for the Examination of Water and Wastewater [36].

For the purpose of assessing the ability for ColiPlate and Colilert to detect very high-risk conditions, the water samples were coded as "1" if the MF method had 100 or more CFU per 100 mL, and densities less than 100 CFU were coded as "0". A simple logistic regression was then performed for predicting very high-risk levels using the $\log_{10}$ ColiPlate results in one model and the $\log_{10}$ Colilert results in another model. Upon running each regression, the discrimination of each method was evaluated in Stata 15 using the area under the receiver operating characteristic (ROC) curve (lroc command) coupled with a sensitivity/specificity assessment using a range of positive predictive values with the lstat command. Model fit was evaluated using the Hosmer–Lemeshow goodness of fit test.

### 3. Results

#### 3.1. Comparison of E. coli Density Results by Enumeration Method

The overall mean $\log_{10}$ *E. coli* density values for the ColiPlate and Colilert methods (Table 1) were not significantly different from the density obtained by the membrane filtration method (according to unpaired *t*-tests with equal variance ($p = 0.541$, $p = 0.505$, respectively)) for the 40 samples. The mean $\log_{10}$ *E. coli* density values were also not different when comparing ColiPlate and Colilert ($p = 0.228$). Furthermore, when assessing *E. coli* method variability dependent upon each sample, paired *t*-test analysis revealed no significant difference between the CFU results obtained by the membrane filtration method with the MPN results obtained by ColiPlate ($p = 0.116$) or Colilert ($p = 0.119$) for the 40 sample pairs. Paired *t*-test analysis comparing ColiPlate and Colilert did reveal a significant difference in the results obtained between these two methods ($p = 0.0160$) for the 40 sample pairs, with ColiPlate indicating higher *E. coli* levels. Visually, the box-and-whisker plot for the three methods also illustrates a significant overlap in the range of results obtained for the same 40 samples (Figure 1). The data are contained within the Supplementary Materials.

**Table 1.** Descriptive statistics describing the distribution of *E. coli* density results obtained from the 40 natural water samples evaluated in this study for method comparison.

| Method | Units | *n* | Mean | SE | Median | Range |
|:------:|:-----:|:---:|:----:|:--:|:------:|:-----:|
| ColiPlate | $\log_{10}$ MPN | 40 | 1.98 | 0.13 | 2.18 | 0.0–3.23 |
| Colilert | $\log_{10}$ MPN | 40 | 1.78 | 0.11 | 1.75 | 0.0–3.01 |
| MF + mTEC [1] | $\log_{10}$ CFU | 40 | 1.88 | 0.11 | 2.01 | 0.2–3.45 |

[1] Membrane filtration method with modified membrane-thermotolerant *E. coli* (mTEC) agar.

Scatterplots illustrating the positive relationship between the MF method with mTEC agar and ColiPlate and Colilert show a strong relationship with each (Figure 2). A visual assessment of the scatterplots relative to these 40 water samples showed a slightly stronger agreement between the MF method and ColiPlate than with Colilert (Figure 2). Overall, Pearson correlation analysis suggested a relatively strong correlation between the results obtained using the three methods. A significant correlation ($R^2 = 0.754$) was observed between the $\log_{10}$ ColiPlate and the $\log_{10}$ MF method results (slope = 1.04, 95% confidence interval: 0.84–1.23). The $R^2$ value between the $\log_{10}$ Colilert and $\log_{10}$ MF method results was 0.679 (slope = 0.85, 95% confidence interval: 0.65–1.04). The correlation between $\log_{10}$ ColiPlate and $\log_{10}$ Colilert was moderate ($R^2 = 0.602$, slope = 0.67, 95% confidence interval: 0.49–0.85), and the y-intercept was significantly greater than 0 (b = 0.45, 95% confidence interval: 0.07–0.83). The three correlations were statistically significant ($p < 0.0001$). For comparison with other studies, Spearman's rank order correlations between the results from ColiPlate and membrane filtration ($\rho = 0.820$) as well as Colilert ($\rho = 0.704$) were

significant ($p < 0.0001$). Similarly, a significant Spearman correlation ($p < 0.0001$) was observed between the membrane filtration and Colilert results ($\rho = 0.785$).

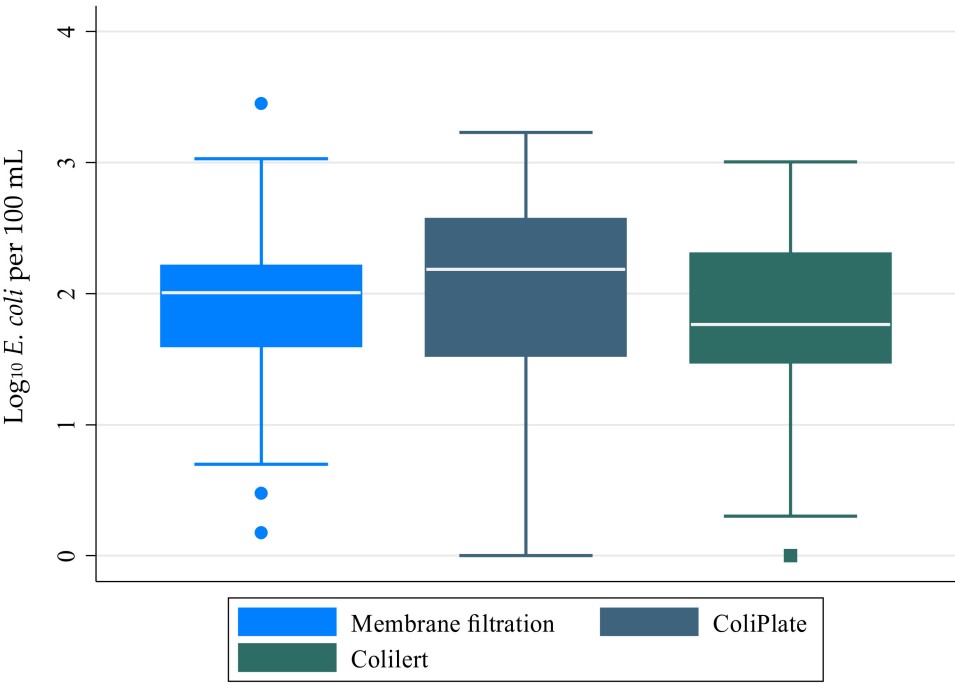

**Figure 1.** Box-and-whisker plot illustrating the determined *E. coli* density distribution for the three methods evaluated in this study whereby each box illustrates the interquartile range and the median, the whisker lines represent the most extreme values within 1.5 times the interquartile range, and the symbols depict outside values.

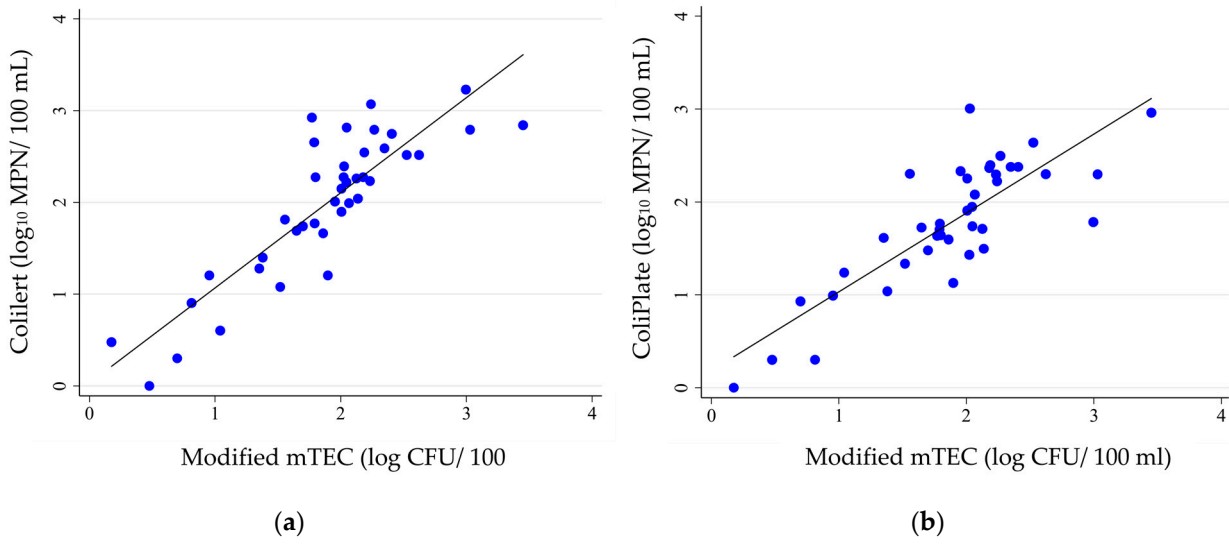

**Figure 2.** Scatterplots with a linear regression line depicting the relationship between the results obtained by the membrane filtration (mTEC agar) method with the results obtained by (**a**) the ColiPlate method and (**b**) the Colilert method.

### 3.2. ColiPlate and Colilert for the Detection of Very High-Risk E. coli Density

Among the 40 samples, a total of 21 (52.5%) exceeded the WHO's very high-risk level of 100 CFU per 100 mL ($\log_{10}$ 2.0 CFU per 100 mL) for drinking water when using the membrane filtration method. The two simple logistic regression analyses that were performed demonstrated very strong associations between an increase in $\log_{10}$ *E. coli*

density measured by each MPN method with the likelihood of observing a very high-risk *E. coli* density determined by the membrane filtration method (Table 2).

**Table 2.** Two simple logistic regression models showing the relationship between the *E. coli* density measurements obtained using ColiPlate and Colilert for predicting a very high-risk *E. coli* density (> 100 colony-forming units per 100 mL) in natural water samples using the membrane filtration method.

| Method/covariate | Units | β [1] | SE β [2] | OR [3] | OR, 95% CI [4] | *p* |
|---|---|---|---|---|---|---|
| ColiPlate | $\log_{10}$ MPN | 3.54 | 1.14 | 34.51 | 3.72–320.49 | 0.002 |
| Constant | | −7.18 | | | | |
| Colilert | $\log_{10}$ MPN | 3.79 | 1.20 | 44.47 | 4.24–465.50 | 0.002 |
| Constant | | −6.75 | | | | |

[1] β: beta coefficient; [2] SE β: standard error of β; [3] OR: odds ratio; [4] CI: confidence interval.

Given the small sample size for each test (*n* = 40), the Hosmer–Lemeshow goodness of fit test was performed using four groups to ensure at least eight observations were in each quantile of the estimated probability of exceeding very high-risk *E. coli* density. The actual observations of exceeding or not exceeding high-risk *E. coli* density in the four quantiles (quartiles) for both the ColiPlate and Colilert models were not significantly different than the modeled expected counts (*p* > 0.05).

The two models (ColiPlate and Colilert) both exhibited excellent discrimination in predicting very high-risk *E. coli* density as ascertained by the membrane filtration method (Figure 3). Specifically, the area under the ROC curve (AUC) was 0.892 and 0.890 for ColiPlate and Colilert, respectively. As a rule of thumb for interpreting ROC curves, areas under the ROC curve ranging from 0.8 to 0.9 are described as providing excellent discrimination [37].

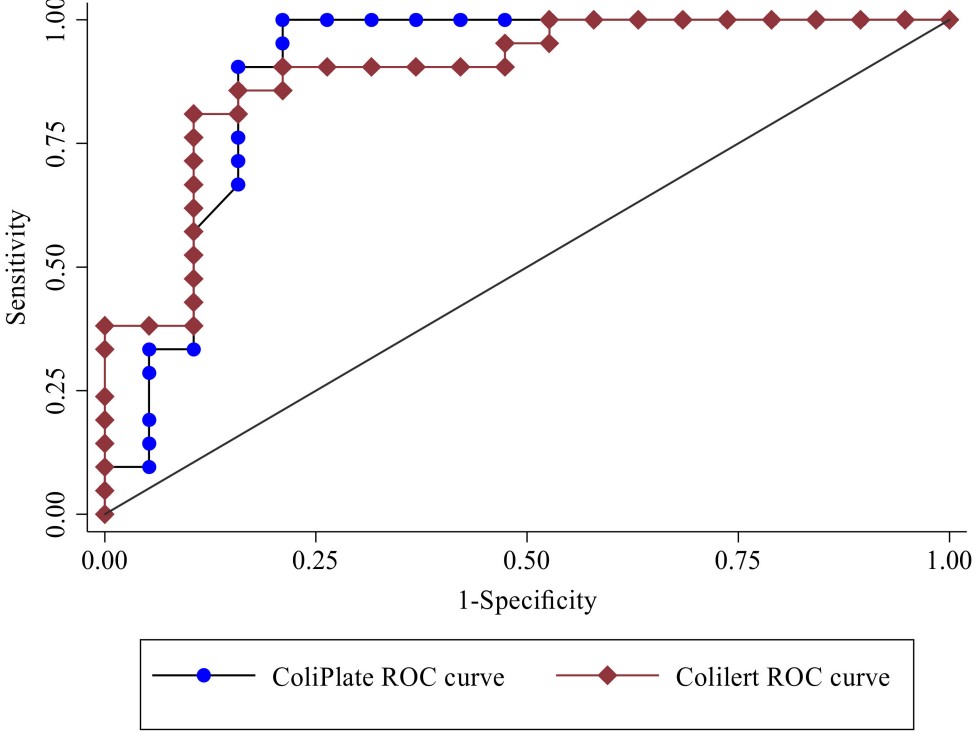

**Figure 3.** Receiver operating characteristic (ROC) curves and the area under the curve (AUC) for the ColiPlate (blue) and Colilert (red) most probable number (MPN) methods used in this study for predicting > 100 colony-forming units of *E. coli* obtained by membrane filtration with mTEC agar.

In examining the ability of each MPN method (ColiPlate and Colilert) to correctly identify the very high-risk *E. coli* density observation by membrane filtration, both tests provided similar correct classification results. Specifically, when using the predicted positive outcome probability (probability cutoff) value of 0.5, which is the probability value typically used in evaluating model discrimination, the ColiPlate results correctly classified 87.5% of samples, and the Colilert results correctly classified 85.0% (Table 3). When using the probability cutoff value of 0.5, the Colilert method had slightly higher specificity and positive predictive values than ColiPlate, whereas ColiPlate had slightly higher sensitivity and negative predictive values (Table 3).

**Table 3.** Performance of the ColiPlate and Colilert logit models for predicting water quality results exceeding the World Health Organization-determined very high-risk level whereby *E. coli* density is or exceeds 100 colony-forming units per 100 mL.

| Method | Probability Cutoff | Sensitivity (%) | Specificity (%) | Positive predictive Value (%) | Negative Predictive Value (%) | Correctly Classified (%) |
|---|---|---|---|---|---|---|
| ColiPlate | 0.40 | 95.00 | 78.95 | 83.33 | 93.75 | 87.50 |
| Colilert | 0.40 | 90.48 | 73.68 | 79.17 | 87.50 | 82.50 |
| ColiPlate | 0.45 | 95.24 | 78.95 | 83.33 | 93.75 | 87.50 |
| Colilert | 0.45 | 85.71 | 84.21 | 85.71 | 84.21 | 85.00 |
| ColiPlate | 0.50 | 90.48 | 84.21 | 86.36 | 88.89 | 87.50 |
| Colilert | 0.50 | 80.95 | 89.47 | 89.47 | 80.95 | 85.00 |
| ColiPlate | 0.55 | 85.71 | 84.21 | 85.71 | 84.21 | 85.00 |
| Colilert | 0.55 | 76.19 | 89.47 | 88.89 | 77.27 | 82.50 |
| ColiPlate | 0.60 | 85.71 | 84.21 | 85.71 | 84.21 | 85.00 |
| Colilert | 0.60 | 76.19 | 89.47 | 88.89 | 77.27 | 82.80 |

## 4. Discussion

### 4.1. Comparison of the Observed Correlations with Related Studies

Peer-reviewed studies evaluating ColiPlate relative to membrane filtration methods are limited. One study [26] using 40 quantifiable natural water samples observed a very strong correlation ($R^2 = 0.954$) between ColiPlate MPN densities, with CFU densities obtained via MF with m-FC-BCIG (membrane fecal coliform agar with 5-bromo-4-chloro-3-indolyl-β-D-glucuronide (BCIG)). In that ColiPlate study [26], the samples came from water treatment plant intakes and wastewater effluents in Ontario, Canada. The same study also evaluated 17 water samples spiked with culture-grown *E. coli* (ATCC 13706) and observed a similarly strong correlation between the MF method and ColiPlate ($R^2 = 0.935$). In this Kentucky study, the strength of association between ColiPlate and MF with modified mTEC agar was strong ($R^2 = 0.754$) but appreciably lower than the value ($R^2 = 0.954$) observed in the study evaluating ColiPlate in natural water [25]. Additionally, the data from this study were log-transformed prior to correlation analysis due to the nonparametric nature of the data.

There are more peer-reviewed studies examining the relationship between Colilert (and Colilert-18) relative to other methods than for ColiPlate. In one study of 234 natural water samples from five Lake Michigan beaches in Wisconsin (US) that compared the Colilert-18 results to the results obtained by MF with mTEC agar, the $R^2$ values across the five beaches were 0.77, 0.89, 0.93, 0.96, and 0.98 [38]. An additional study that compared Colilert to an MF method in Virginia streams and rivers near agricultural areas observed a comparable strength of correlation as the Lake Michigan studies ($R^2 = 0.914$) [39]. The results obtained in this study had a Colilert correlation value ($R^2 = 0.679$) appreciably lower than the values observed in the Lake Michigan Colilert-18 beach studies ($R^2 = 0.77$–0.98) and the Virginia stream/river Colilert study ($R^2 = 0.914$) [38,39]. One study of Tennessee (US) streams and rivers compared the mTEC agar and Colilert results from 53 samples and observed a weaker correlation ($R^2 = 0.67$) than that observed in other studies [40]. Again, it is important to note that the $R^2$ values in these comparison studies were derived from *E. coli* results that had not been log-transformed, whereas this Kentucky study used log-transformed data.

The potential reasons for lower correlation values among the methods in this study relative to other studies likely relate to several factors such as (1) the number of analysts performing the methods across the water samples, (2) the limited experience of some student analysts in their ability to consistently implement each method protocol for their individual sample relative to other students, (3) the variation in individuals interpreting positive versus negative *E. coli* CFU or wells, and (4) the substantial environmental and, presumably, microbial community variability at the sample collection sites. Additionally, several of the water samples had *E. coli* density that exceeded the optimal number of CFU (>80 CFU) to be counted on modified mTEC agar plates [20]. A key source of error or variation could also relate to the mixing protocols followed by each student investigator. Mixing intensity (number of times and degree of vigorousness) of a sample prior to each method (and during the procedure in the case of Colilert) may have contributed to variability in the results by method types.

*4.2. Application of Results to Water Quality Analyses in Low-Resource Environments*

The research comparing inexpensive growth medium Aquatest with Colilert and other methods observed an association between AT and Colilert comparable to the association observed in this study between ColiPlate and Colilert [31]. Specifically, Aquatest had strong correlations with Colilert-18 in temperate waters in the United Kingdom ($\rho = 0.802$) and Switzerland ($R^2 = 0.960$, $\rho = 0.950$), as well as in subtropical waters in South Africa ($\rho = 0.793$) [41,42]. Although the Aquatest medium was not used in this study, we can see for comparison that ColiPlate had a similar correlation with the membrane filtration mTEC agar method ($\rho = 0.820$) but a slightly weaker correlation with Colilert ($\rho = 0.704$).

For additional analysis, an examination of sensitivity and specificity of a method can take place. An analysis by Bain et al. [41] observed a higher amount of sensitivity for the AT medium relative to Colilert (98% versus 87%) in correctly detecting a higher frequency of true *E. coli*-positive samples among all the samples. The false negative rate which directly reduces sensitivity ranged for Colilert from 7.6% to 18.9% in the Caribbean (Trinidad and Tobago) [42], to 27.3% in Florida [43], and to as high as 36.4% in tropical waters in southern China [44]. As long as some (10–20%) *E. coli* are presumably MUG-negative [44–46], they may go undetected by Colilert, but MUG-negative *E. coli* would also go undetected with the AT-MUG medium and ColiPlate which also use MUG. The research on the specificity and sensitivity of ColiPlate is limited and they are not described further than the assessments of density relative to one MF method [25].

According to the comparison of the AT medium to Colilert in previous studies [41,42], the specificity (the frequency of detecting true negatives among all samples) was similar for both methods (~95%). In this study, unlike in the study by Bain et al., the sensitivity and specificity were evaluated on the basis of whether or not the results of the ColiPlate and Colilert methods (in MPN per 100 mL) exceeded 100 (in *E. coli* CFU per 100 mL) as obtained by the membrane filtration method with modified mTEC agar. The decision for this evaluation approach to take place using this *E. coli* density cutoff was two-part: (1) true negatives (0 CFU per 100 mL) were not observed with the reference method for these environmental samples; (2) in many areas around the world with human habitation, untreated natural waters used for drinking and contact recreation or bathing are often contaminated by fecal indicator bacteria which results in *E. coli* being present. For these reasons, densities exceeding 100 CFU per 100 mL represented the positive condition. The cutoff of 100 CFU per 100 mL was selected as it represents the WHO decisionmaking threshold declaring the water is likely to present a very high risk for drinking water users [43].

Using the default probability threshold of 0.50 (Table 3) which represents a 50% probability of a sample having 100 CFU per 100 mL, ColiPlate had greater sensitivity (90%) than Colilert (81%), whereas Colilert had greater specificity (89%) than ColiPlate (84%). Overall, the frequency of correct classification of the 40 samples was essentially the same (chi-squared test, $p = 0.745$), with 35 (87.5%) correct by ColiPlate, 34 (85%)—by Colilert.

These results are corroborated by the excellent discrimination of the two logit models both having an area under the ROC curve of 89%. Based upon these findings and the relative ease of using ColiPlate in resource-limited conditions, there is evidence that the field-suitable ColiPlate kit method represents a meaningful approach for gathering data for research purposes, evaluating interventions, and potentially determining the presence of waterborne disease risks associated with elevated levels of fecal indicator bacteria.

### 4.3. ColiPlate E. coli Densities Greater Than Densities from Membrane Filtration and Colilert

Overall, the mean, median, and upper ranges of the *E. coli* density observed by the ColiPlate method were higher than the density determined by the MF method and Colilert (Table 1). The Colilert density in the paired *t*-test analysis being significantly lower than that observed with ColiPlate may have been due to Colilert having greater specificity (having less false positives in the MPN density determination by better preventing non-target (non-*E. coli*) bacteria from glowing). Greater specificity could be possible due to the use of any inhibitors in the Colilert medium not included in the ColiPlate medium and/or more effective sealing of wells in the Colilert/Quanti-Tray system preventing spillage of UV-fluorescent compounds from *E. coli*-positive wells into *E. coli*-negative wells during walking/transport from the incubator to the UV viewing cabinet. Additionally, ColiPlate may have greater recovery (greater survival of *E. coli*) or greater sensitivity (ability to correctly classify true positive wells for detecting *E. coli*). Additional research examining the positive wells for each method for the true presence or absence of *E. coli* would be needed to confirm these sensitivity and specificity speculations.

In the initial publication regarding ColiPlate [26], the *E. coli* density obtained with ColiPlate was 168% higher than that observed using m-ENDO-LES agar plates for *E. coli* enumeration following MF. Several plausible explanations were presented; however, the most emphasized explanation was that the method resulted in a greater recovery of injured or weakened cells than an MF method [26], which was supported by the prior studies comparing recovery between multiple tube fermentation (MTF) methods and MF methods [47,48]. Similar to the initial ColiPlate study, we observed a higher *E. coli* density with ColiPlate, although not to the same degree. In this study, using the *E. coli* density without $log_{10}$ transformation, the mean and median ColiPlate results were 26% and 48% higher than when the MF method was used, respectively. Despite being higher, the results of both methods were not significantly different in the paired *t*-test analysis using the $log_{10}$-transformed data, nor were the y-intercepts from unity (0).

The information available at the time of the initial ColiPlate publication in 1997 included information received by the Ontario Ministry of Environment and Energy in Canada suggesting that Colilert, another method built upon MTF methods, should also have a greater *E. coli* detection frequency than MF methods [26]. However, conversely to that speculation, our mean and median Colilert density results were 21% and 45% lower than the MF method results, respectively. Despite being lower, the results were not significantly different in the paired *t*-test analysis of the $log_{10}$-transformed data. These findings of fewer *E. coli* by Colilert versus MF were observed in two studies comparing Colilert to MF methods in the same temperate zone as this study, of which one (Virginia, United States) had similar land use attributes as this study (agriculturally impacted environment with some forested landscape and low housing density) [39]. In that study, *E. coli* density was obtained for 396 samples across 10 different locations, and for all the 10 locations, the mean results were greater for the MF method than for Colilert. The MF method comparisons between studies are limited though in that the growth medium used in the MF method in this study was different from the media used in the Virginia study. In a more recent study (2019), the relationship between an MF method using mTEC agar with Idexx Colilert-18 across 70 temperate water samples from Switzerland observed a lower *E. coli* density with Colilert-18 than with the MF method [42]. The slope (b = 0.85, 95% confidence interval: 0.74–0.96) of the regression line in the Swiss water samples that included the Colilert-18 and

MF results was nearly identical to the one observed in this study (b = 0.85, 95% confidence interval: 0.66–1.04).

*4.4. Study Limitations and Challenges Presented*

While the study demonstrates strong correlations between the various methods, the correlations observed in this study are not as high as observed between the various methods or approaches in other studies (described in Section 4.1). Beyond the potential differences attributable to growth media and diversity of *E. coli* strains or microbiomes from the variety of environments sampled, some variability could be attributable to the results being obtained by persons not accustomed to routine analysis using these methods. The variability attributable to multiple novice analysts is speculative as the study has two apparent weaknesses: (1) the study design did not incorporate and utilize a consistent professional analyst to simultaneously analyze split water samples which would have enabled paired analyses and assessments of variability in the results collected by the group of novice analysts relative to a consistent professional; (2) the approach also lacked replicates for the *E. coli* enumeration methods which did not allow for an assessment of inter- and intrasample baseline variability. Despite these limitations, these data add to the evidence that both ColiPlate and Colilert, when used by novice scientists, can effectively discriminate high-risk conditions for water-related illnesses from untreated drinking water. Greater repetition of all the methods, particularly with regard to sample container mixing practices, would likely improve consistency and the strength of correlations.

Another area of improvement in assessing correlations would be having greater consistency by analysts who may periodically discern positive/negative events from the ColiPlate and Colilert kits when there are marginal color changes and/or marginal glowing of wells that are difficult to discern even when using manufacturer-provided control trays and/or images. Similarly, unusual appearances and colors and/or clustering of colony-forming units on modified mTEC agar plates can also present inconsistencies in results across various inexperienced analysts.

An additional challenge presented in this study (and the related studies) includes the potential for variation in natural and/or pollutant-related inhibitors that could alter or prevent reactions for one method while having different or no impact on the comparison methods. Additionally, chemicals or agents in the water may also mask glowing coloration in DST methods [49], which could be possible in the waters from this study given their regular impairment by nutrients and eutrophication. Furthermore, various aquatic microbiomes may be more likely to enable false-positive environmental flora to present [42,44,45,50]. This study did not evaluate the true/false positive or true/false negative rates of the various methods; however, there is literature documenting that these rates can be variable and sometimes problematic based upon water temperature among other potential factors [42,44,45,50].

Lastly, the sampling methodology used in this study does not enable broad extrapolation of the results to all aquatic environments used for drinking water. All the samples were from relatively proximal locations in Kentucky, and none of these samples were planned for drinking water use as could occur with similarly situated waterbodies in low-resource settings of the world. Furthermore, the ColiPlate results were obtained in tandem alongside Colilert with the same conditions (35 °C for 24 h). In low-resource settings, these incubation conditions may be maintained with specialized field incubation approaches and/or improvisation.

## 5. Conclusions

- Strong correlations were observed between the *E. coli* densities obtained by ColiPlate and Colilert with the MF method using modified mTEC agar.
- The ColiPlate and Colilert methods provided excellent discrimination in properly classifying the WHO high-risk conditions for drinking water-related illnesses.

- ColiPlate had higher sensitivity but lower specificity than Colilert in this study for determining the WHO high-risk conditions for drinking water-related illnesses.
- The data from these Kentucky (USA) water samples suggest ColiPlate continues to effectively assess fecal indicator bacteria in natural water relative to Colilert, and the evidence suggests that ColiPlate could potentially be an effective tool for citizen scientists evaluating untreated drinking waters and natural waters.

**Supplementary Materials:** The following are available online at https://www.mdpi.com/article/10.3390/w13131804/s1, Table S1: Data Availability_Water.

**Author Contributions:** Conceptualization, J.W.M.; methodology, J.W.M. and C.J.G.; formal analysis, J.W.M. and C.J.G.; investigation, J.W.M., C.J.G., and A.K.M.; resources, J.W.M.; data curation, C.J.G. and A.K.M.; writing—original draft preparation, C.J.G.; writing—review and editing, J.W.M. and C.J.G.; visualization, J.W.M.; supervision, J.W.M.; project administration, J.W.M. All authors have read and agreed to the published version of the manuscript.

**Funding:** This research received no external funding. The supplies used in this study were purchased with funds from the Master of Public Health program and the Department of Environmental Health Science and Medical Laboratory Science at Eastern Kentucky University (EKU). Dissemination support was provided by the Master of Public Health program at EKU.

**Institutional Review Board Statement:** Not applicable.

**Informed Consent Statement:** Not applicable.

**Data Availability Statement:** The data are contained within the Supplementary Materials. The data presented in this study are available in [DataAvailability_Water.xlsx].

**Conflicts of Interest:** C.J.G. and A.K.M. declare no conflicts of interest. J.W.M. declares being the inventor of a competitive product, ColiGlow (PCT/US2021/18679). Data collection and analysis for this study by C.J.G. and A.K.M. occurred with J.W.M. in 2016 and 2017. The funder of the supplies, Eastern Kentucky University, had no role in the design of the study; in the collection, analyses, or interpretation of data; in the writing of the manuscript, or in the decision to publish the results. The content provided here represents the original findings of the authors and does not necessarily represent the views or interpretations of Eastern Kentucky University.

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
