# Peer review of "Comparison of the ColiPlate™ Kit with Two Common E. coli Enumeration Methods for Water"

_water, doi:10.3390/w13131804_

Round 1
Reviewer 1 Report
This paper compares three different E. coli assays, using 40 environmental water samples having a range of E. coli levels. The statistical analysis is clear and appropriate.
One weakness is that no replicates were conducted, so it’s not possible to quantify the baseline variability of any of the assays for a given sample. This isn’t a major concern, but might be noted in the limitations.
Most of the statistical analysis is set up to compare each of the two MPN methods to the MF method; in effect the MF is being considered as the reference method. This would be good to explain earlier in the manuscript. Only after the ROC analysis and logistic modelling, when there are phrases like “correctly identify” and “correct classification results” is it clear that the MF is being taken as the reference method. Is there any justification for considering MF as more “correct” than the Colilert method, which is also EPA certified?
I think it is correct and justified to make most of the statistical analysis on log-transformed data. But I’m not sure about the language justifying this: line 122 reads “given the nonparametric distribution of the E. coli data… data were normalized using a log transformation” and lines 185 and 247 again refers to the nonparametric nature of the data. I may be mistaken, but I think data, and data distributions, are not either parametric or non-parametric. Analytical methods are parametric and non-parametric, and some of the methods used in the paper (e.g. Spearman correlations) are indeed non-parametric methods. A transformation doesn’t change the analysis to non-parametric – for example the logistic regression made on log-transformed data is still a parametric analysis. I think the log distribution is justified because the distribution of E. coli is skewed, and it seems that the distribution is more manageable after the log-transform. But it is still parametric.
The authors spend I think too much time in the conclusion on possible reasons that the correlations found in this study are lower than others reviewed in the literature: there’s a lot about the limitations of testing being done by university students rather than professional lab technicians (lines 264-275, and later 348-372). But it seems quite plausible that the main cause is that this study made a log transformation, which will of course reduce the influence of the samples with the highest counts, which might dominate the statistical analysis of the non-transformed data. The authors note this difference (e.g. lines 262-263) in passing but don’t come back to it in the (lengthy) discussion on reasons for lower correlations. Consider checking what the correlation coefficients would be on the non-transformed data as a kind of sensitivity check.
Most of the analysis is about correlation, but there is some interesting analysis about recovery. Of course two methods could be extremely highly correlated but have very different recoveries, and it does seem that Coliplate has higher recovery than Colilert. Again if MF is taken as the reference method, consider a few more parametric analyses of recovery. The authors indicate the differences of mean and median, how about the slope of a non-transformed linear regression curve? Or the intercept of a log-transformed linear regression curve (if there is no significant intercept for the linear regression)?
The closing sentence (lines 380-381) are not warranted: the authors suggest that the “flexibility on incubation temperatures and durations” that ColiPlate offers may be an advantage of this method. However, this flexibility has not been described in the text, and has not been examined empirically. Unfortunately, sometimes manufacturer claims of consistent performance over wide ranges of temperature incubations are overrated.
In the discussion in section 4.2 of the Aquatest media, consider citing Genter et al (DOI: 10.1039/c9ew00138g) which also examined the AT media against Colilert in environmental waters from Switzerland and Uganda.
Finally, the writing is generally very good but there are a few typos (lines 290, 319, and 325) and E. coli is not consistently italicized as it should be throughout.
Reviewer 2 Report
The manuscript "Comparison of the ColiPlate™ Kit with Two Common E. coli 2 Enumeration Methods for Water" presents the results of a study conducted to investigate and compare two usual approaches for quantifying E. coli densities in natural water.
The paper is well written and the results are clearly presented.
I suggest the acceptance of this paper after minor revision (see atachment).

Author Response
REVIEWER #2:
The manuscript "Comparison of the ColiPlate™ Kit with Two Common E. coli 2 Enumeration Methods for Water" presents the results of a study conducted to investigate and compare two usual approaches for quantifying E. coli densities in natural water.
The paper is well written and the results are clearly presented.
I suggest the acceptance of this paper after minor revision (see attachment).
RESPONSE: Thank you for your review of the manuscript. We have made an effort to improve communication in the paper on the discrepancy between 41 samples and 40 samples. The abstract now states 41 samples. The methods now state as follows:
Among the 41 water samples, one sample exceeded the range of all tests and was excluded. Summary statistics (mean, median, and range) were obtained for the 40 remaining…
Reviewer 3 Report
The work focuses on the comparison of three methods for the evaluation of E. coli in natural waters.
To compare the methods, 40 samples were collected from natural water sources in Kentucky (USA).
The methods are: membrane filtration in combination with m-TEC agar, Idexx Quanti-Tray® 2000 with Colilert® substrate and the Bluewater Biosciences ColiPlate Kit.
The work is well described, however some aspects in the introduction part and in the methods should be revised and improved:
- Lines 31-33: Authors should add bibliographic references relating to the use of E. coli as an indicator in recreational waters (eg.: An YJ, et al. Monitoring E. coli and total coliforms in natural spring water as related to recreational mountain areas. Environ Monit Assess. 2005 Mar; 102 (1-3): 131-7. doi: 10.1007 / s10661-005-4691-9; Giampaoli S et al. Health and safety in recreational waters. Bull World Health Organ. 2014 Feb 1; 92 (2): 79. Doi: 10.2471 / BLT.13.126391).
- Lines 67: the authors state that the Bluewater Biosciences ColiPlate. is similar to the Indexx kit, it should be clarified in what they are similar and what are the differences, for example the type of substrate contained, the reactions.
- Lines 77-147: authors should divide materials and methods into subparagraphs to allow the reader to better follow the different technical issues. Furthermore, the two MPN methods should be explained better and in detail, especially on the interpretation of the results.
- Lines 80: the authors should better specify the type of water used for the study, describing in detail the different types and sampling. This should be discussed in light of the weak/unclear results as outlined in the discussion.
- Lines 361-369: the discussion of the results should also consider other kind of waters and the application of these rapid methods in different context. Authors may want to explain better the differences in results based on the kind of sampled water. Indeed, different water sources may impact on the effectiveness of the results (e.g. contaminants, inhibitors, confounders, storage and transport conditions, etc.. ).
Reviewer 4 Report
This manuscript reports an investigation of a given E. coli. testing method compared to the other two enumeration methods. The topic is meaningful. The experiments were carefully carried out. The manuscript was prepared well with clear elucidation and is generally good. However, I have some minor comments shown below.
General comments:
Not all readers are familiar with and/or have used the ColiPlate Kit. Please give more detailed background information about the ColiPlate. Does it have the same substances as other methods use? What is the benefit of using it? Please also provide important information that is listed on the specifications of this product.
Specific comments:
Lines 67 to 68: Provide detailed information about ColiPlate, such as how it works. Please indicate the difference between ColiPlate and Idexx Colilert.
Line 76: In this paragraph, please give more detailed discussion about the difference between ColiPlate from others. This should be a reason driving you to test it. In fact, it is not necessary to test it if they were the same thing but with different commercial names.
Line 122: Explain what the nonparametric distribution means and any associated consequence.
Line 154: Expand the discussion and indicate if the significant difference matters.
Line 156: Explain why box-and-whiskers plot showed overlapped results but t-test indicated significantly different.
Line 199: It is necessary somewhere to explain the parameters shown on the first row of Table 2.
Lines 261 to 263: Indicate what results the log-transformation could lead to. Why did you emphasize that you used log-transformed data?
Line 264: Some systematic errors could be mitigated by a better design. I am curious that if a professional person performed the tests, how good the correlation could be. It deserved to be tested.
Line 285: Please show the definitions of sensitivity and specificity explicitly.
Line 286: Here you only mentioned the study of Bain et al. Did any other studies conduct a similar analysis? Please give an overview. The readers need to have a whole picture.
Lines 302 to 304: Discuss the reason why they were different.
Line 306: Explain the x2p term.
Line 309: You indicated ColiPlate could be used in resource-limited conditions. Somewhere in the article, please highlight the benefit of using ColiPlate, ie., how easily it can be used.
Lines 314 to 315: Show quantitative results.
Lines 328 to 331: The true values with 26% and 48% in difference are high. Taking a logarithm does not reflect the reality. In addition, do you have any suggestions on increasing the accuracy?
Lines 335 to 337, “Conversely to that speculation … were 21% and 45% lower than the MF method results”: Discuss the possible reason why your results were different.
Line 339, “one study”: How about other studies? Did they agree or not? Please give an overview of all relevant studies.
Lines 348 to 349: Specify quantitatively what other studies showed about the correlation.
